# Flavin Adenine Dinucleotide Fluorescence as an Early Marker of Mitochondrial Impairment During Brain Hypoxia

**DOI:** 10.3390/ijms21113977

**Published:** 2020-06-01

**Authors:** Nikolaus Berndt, Richard Kovács, Jörg Rösner, Iwona Wallach, Jens P. Dreier, Agustin Liotta

**Affiliations:** 1Institute for Imaging Science and Computational Modelling in Cardiovascular Medicine Charité—Universitätsmedizin Berlin, Corporate Member of Freie Universität Berlin, Humboldt-Universität zu Berlin and Berlin Institute of Health, 10117 Berlin, Germany; nikolaus.berndt@charite.de; 2Institute for Neurophysiology, Charité—Universitätsmedizin Berlin, Corporate Member of Freie Universität Berlin, Humboldt-Universität zu Berlin and Berlin Institute of Health, 10117 Berlin, Germany; richard.kovacs@charite.de; 3Neuroscience Research Center, Charité—Universitätsmedizin Berlin, Corporate Member of Freie Universität Berlin, Humboldt-Universität zu Berlin and Berlin Institute of Health, 10117 Berlin, Germany; joerg.roesner@charite.de; 4Institute of Biochemistry, Charité—Universitätsmedizin Berlin, Corporate Member of Freie Universität Berlin, Humboldt-Universität zu Berlin and Berlin Institute of Health, 10117 Berlin, Germany; iwona.wallach@charite.de; 5Center for Stroke Research Berlin, Center for Stroke Research Berlin, Charité—Universitätsmedizin Berlin, Corporate Member of Freie Universität Berlin, Humboldt-Universität zu Berlin, and Berlin Institute of Health, 10117 Berlin, Germany; jens.dreier@charite.de; 6Department of Neurology, Charité—Universitätsmedizin Berlin, Corporate Member of Freie Universität Berlin, Humboldt-Universität zu Berlin, and Berlin Institute of Health, 10117 Berlin, Germany; 7Department of Experimental Neurology, Charité—Universitätsmedizin Berlin, Corporate Member of Freie Universität Berlin, Humboldt-Universität zu Berlin, and Berlin Institute of Health, 10117 Berlin, Germany; 8Bernstein Center for Computational Neuroscience Berlin, 10117 Berlin, Germany; 9Einstein Center for Neurosciences Berlin, 10117 Berlin, Germany; 10Berlin Institute of Health, Charité—Universitätsmedizin Berlin, Corporate Member of Freie Universität Berlin, Humboldt-Universität zu Berlin and Berlin Institute of Health, 10117 Berlin, Germany; 11Department of Anesthesiology and Intensive Care, Charité—Universitätsmedizin Berlin, Corporate Member of Freie Universität Berlin, Humboldt-Universität zu Berlin and Berlin Institute of Health, 10117 Berlin, Germany

**Keywords:** FAD, brain, hypoxia, spreading depolarization, mitochondria, computational modeling

## Abstract

Multimodal continuous bedside monitoring is increasingly recognized as a promising option for early treatment stratification in patients at risk for ischemia during neurocritical care. Modalities used at present are, for example, oxygen availability and subdural electrocorticography. The assessment of mitochondrial function could be an interesting complement to these modalities. For instance, flavin adenine dinucleotide (FAD) fluorescence permits direct insight into the mitochondrial redox state. Therefore, we explored the possibility of using FAD fluorometry to monitor consequences of hypoxia in brain tissue in vitro and in vivo. By combining experimental results with computational modeling, we identified the potential source responsible for the fluorescence signal and gained insight into the hypoxia-associated metabolic changes in neuronal energy metabolism. In vitro, hypoxia was characterized by a reductive shift of FAD, impairment of synaptic transmission and increasing interstitial potassium [K^+^]_o_. Computer simulations predicted FAD changes to originate from the citric acid cycle enzyme α-ketoglutarate dehydrogenase and pyruvate dehydrogenase. In vivo, the FAD signal during early hypoxia displayed a reductive shift followed by a short oxidation associated with terminal spreading depolarization. In silico, initial tissue hypoxia followed by a transient re-oxygenation phase due to glucose depletion might explain FAD dynamics in vivo. Our work suggests that FAD fluorescence could be readily used to monitor mitochondrial function during hypoxia and represents a potential diagnostic tool to differentiate underlying metabolic processes for complementation of multimodal brain monitoring.

## 1. Introduction

In patients with acute brain injury due to stroke, bleeding or trauma, detecting the development of secondary brain injury is challenging during neurocritical care since neurological examination is often limited by unconsciousness [1]. Therefore, invasive neuromonitoring is increasingly performed in patients with severe brain insults [2]. Particularly after ischemia, avoiding further progression of neuronal loss remains the main goal of treatment [3]. The precise moment when brain injury becomes irreversible is difficult to determine even in experimental settings but monitoring alterations in cell physiology and homeostasis might help physicians to understand and treat damage progression [2]. A key phenomenon in developing neuronal insult is the spreading depolarization (SD) continuum, as SD is the mechanism initiating the cytotoxic edema [4,5,6]. While extreme metabolic compromise in the ischemic core results in terminal SD characterized by a high-amplitude negative ultraslow potential [7,8], the penumbra and adjacent tissue typically display a cluster of SDs that are progressively shorter-lasting in the centrifugal direction. Though penumbral SDs are often reversible, they strain the neuronal metabolism, e.g., through cytoplasmic calcium overload [9], and may lead to reductions in tissue oxygen (O_2_), glucose and adenosine triphosphate (ATP) [10,11,12] due to increased energy demand and inverse hemodynamic responses [13]. In the course of these alterations, there are also significant mitochondrial changes [14]. Thereby ATP production necessary for basal metabolism and structural integrity is further impaired [15,16]. Furthermore, mitochondrial impairment and segmentation after ischemia triggers apoptosis via cytochrome c release in the cytosol [17].

Established invasive monitoring procedures in neurocritical care include: monitoring intracranial pressure (ICP), electrocorticography (ECoG), tissue oxygenation (pO_2_) and changes in metabolites and neurotransmitters measured with microdialysis (i.e., lactate, pyruvate, glucose, pyruvate or ions) [18,19]. All these technologies monitor tissue alterations providing valuable but, if at all, indirect information about acute mitochondrial dysfunction and energy compromise, the actual threat during severe brain injury [17].

The fluorescence emission of FAD depends on its redox state and mainly corresponds to the redox state of mitochondrial enzyme complexes [15,20,21]. In the oxidized state, FAD contains an isoalloxazine that emits fluorescence at a ~515 nm wavelength when excited with blue light (i.e., ~460 nm) while the reduced molecule (FADH_2_) does not generate fluorescence [22]. Therefore, useful insights into the mitochondrial redox state can be gained by monitoring autofluorescence at the appropriate wavelengths [23]. Indeed, FAD fluorometry independently or in combination with fluorescence measurements of nicotinamide adenine dinucleotide (NADH) has been performed in vitro and in vivo to gain information on oxidative energy metabolism [15,22,24,25,26,27,28,29,30,31,32,33]. In the last decades, several milestones have been accomplished in the field of NADH imaging in vivo in the brain [34]. Bedside systems for monitoring the NADH redox state in the brain of patients during neurosurgery and for monitoring systemic metabolic dysfunction during hemorrhage and resuscitation have been successfully established [35,36,37]. Compared to NADH, FAD measurements are more difficult due to less intense autofluorescence but excitation is less phototoxic, as the wave length for excitation is longer [22,33]. Further disadvantages of NADH measurements result from the facts that NADH redox changes not only occur in the mitochondria but also in the cytoplasmic compartment and in erythrocytes so that the signal is modified by changes in regional cerebral blood flow (rCBF) [38]. By contrast, changes in FAD fluorescence specifically originate from the pyruvate dehydrogenase (PDHC), the α-ketoglutarate dehydrogenase (KGDHC), the glycerol-3-phosphate dehydrogenase (G3PDH) and the succinate dehydrogenase (SUCCDH) complexes. Thus, using computer models, specific changes of intramitochondrial redox potential can be estimated based on FAD fluorescence [21,39,40]. Computer simulations of cellular metabolism fitted with experimental data then permit the study of changes in metabolites and the rates of their mutual chemical interconversion in response to varying external conditions [32,39,40,41].

During ischemia, FAD fluorescence decay has been previously measured in rat liver in vivo [42]. Here, we studied FAD fluorescence changes during O_2_ decrease in neocortical rat brain slices while monitoring synaptic transmission, tissue pO_2_ and the extracellular potassium concentration ([K^+^]_o_). Integrating experimental data with computational modeling, we reconstructed specific functional changes in the oxidative phosphorylation. Furthermore, we established FAD in vivo measurements in rats during terminal hypoxia and then reconstructed possible enzymatic scenarios in silico. As a long-term goal, we suggest that assessing mitochondrial function with FAD imaging could help clinicians to better understand the progression of neuronal damage and could serve as a tool for treatment stratification during neurocritical care.

## 2. Results

### 2.1. Oxygen Depletion Correlates with Changes in FAD Redox State and Impairment of Synaptic Transmission in Brain Slices

In vitro brain slice preparations allow full control of oxygenation while the changes in tissue pO_2_, tissue excitability and FAD fluorescence are simultaneously monitored. To assess changes in FAD fluorescence from baseline during mild hypoxia, we lowered oxygenation from 95% to 50% for 20 min (Figure 1). In control conditions (i.e., 95% O_2_), pO_2_ baseline was 91.43 mmHg (interquartile range: 81.62, 108.9) and decreased to 14.7 mmHg (9.9, 32.83) when 50% O_2_ was applied (depth of ~50 µm from slice surface, *n* = 9, *p* < 0.001). During low O_2_, a slight but significant increase in baseline [K^+^]_o_ from 3.0 mM (3.0, 3.1) in the control to 3.2 mM (3.3, 3.6) occurred (*n* = 9, *p* < 0.001). In all experiments, we observed a simultaneous decrease in FAD fluorescence (Δf/f_0_) by ~20% from 1.01 (1.0, 1.02) to 0.84 (0.82, 0.83) indicating a reductive shift of the aforementioned mitochondrial complexes during moderate hypoxia (*n* = 9, *p* < 0.001).

Changes in synaptic transmission during control and hypoxia were assessed by measuring the amplitude of stimulus-induced population spikes (see Methods and Figure 2). The amplitude of population spikes significantly decreased from 1.58 mV (1.28, 3.21) to 0.35 mV (0.19, 0.45) indicating impairment of synaptic transmission during moderate hypoxia (*n* = 9, *p* = 0.004).

To better characterize FAD fluorescence changes, we generated a sudden increase in energy demand by applying a 20 Hz/2 s stimulation train during each condition (Figure 2). This kind of stimulus typically generates a biphasic redox change in the tissue, i.e., a first oxidative peak followed by a reductive undershoot in FAD signal [29,33,43]. We measured a consistent decrease in FAD peak amplitude from 2.32 (1.79, 3.38) to 1.06 (0.73, 1.54) and in FAD undershoot from 3.13 (1.97, 3.84) to 1.90 (1.45, 2.31) when slices were treated with 95% and 50% oxygen (data concerning changes in Δf/f_0_, *n* = 9, *p* < 0.001 and *p* = 0.006 respectively). In line with a state of energy deficiency, simultaneously recorded [K^+^]_o_ increases and pO_2_ dips decreased, as tissue excitability was impaired (Figure 1A and Figure 2B). Since redox changes in several mitochondrial enzymes might contribute to the observed alteration in FAD fluorescence, we used computer simulations within our experimental conditions to uncover the main source of signal.

### 2.2. Slice Modeling: The Influence of Oxygen Supply on the Basal and the Activation-Induced FAD Response

We modeled hypoxia-associated FAD fluorescence changes by using a computational model where each horizontal layer of the slice (down from the surface) was treated as a homogenous entity coupled by diffusive O_2_ transport (see Appendix A). Homologous to the in vitro situation, available glucose concentration was constant and sufficient in any layer of the slice (equilibrium condition due to constant bath perfusion of artificial cerebrospinal fluid (aCSF) with 10 mM glucose). Metabolic capacities (i.e., the abundance of metabolic proteins) as well as basal and maximal ATP-consumption rates were assumed to be identical in all layers of the slice. The FAD fluorescence signal was assumed to be the integral response of all layers. In line with the experimental data (as approximately recorded in submerged conditions with 8 mL/min flow), we put the pO_2_ at the slice surface to 350 mmHg at 95% oxygenation and 150 mmHg at 50% oxygenation.

Following the experimental protocol, we simulated changes in baseline and during short periods of neuronal activation (stimulation trains) as depicted in Figure 3 and Figure 4. The first stationary metabolic state of the slice model corresponded to 95% oxygenation. Although there was a steep decline in O_2_ concentration from the surface to the core of the slice, the pO_2_ greatly exceeded the oxygen affinity of complex IV of the respiratory chain (Km = 3 mmHg) in almost all layers. Hence, the metabolic activity including the O_2_ consumption rate and the FAD reduction states were almost identical in all layers. Only the layers closest to the core, where pO_2_ decreased below 10 mmHg exhibited a higher FAD reduction state. A first electrical stimulation (2 s) during the control condition was simulated by a steep increase in ATP demand and a corresponding influx of calcium (details in Figure 4) [32,39]. This sudden increase in energy demand led to a drop in ATP, stimulating the respiratory chain and the oxidative phosphorylation and thereby increasing O_2_ consumption rate [32]. The increased O_2_ consumption led to activation of the citric acid cycle together with an oxidative shift in its FAD containing enzymes, as well as a drop in pO_2_ throughout the slice. Once the ATP demand decreased to normal values, the persisting activation of the citric acid cycle resulted in a reductive shift in the FAD containing enzymes, while the O_2_ consumption rate and the pO_2_ throughout the slice returned to unstimulated values. As long as pO_2_ in the different layers of the slice was high enough to keep complex IV of the respiratory chain saturated, the metabolic response throughout the slice remained homogenous. However, as can be seen in Figure 3, even at 95% oxygenation, the lowest layer of the slice might experience desaturation of complex IV and therefore an increased reductive shift during electrical stimulation.

The situation changed when oxygenation was decreased to 50% (phase 3). The O_2_ supply was now insufficient to ensure saturation of complex IV in all layers, as the lower layers of the slice experienced increasingly severe hypoxia. While the top layers of the slice were unaffected, the lower layers responded to the increasing hypoxia with an increased FAD reduction. The lower the local pO_2_ was, the lower the O_2_ consumption rate became, and the higher the FAD reduction state was, as it was not oxidized anymore by the respiratory chain. At this point, the slice did not exhibit a homogenous metabolic state anymore. Assuming that the experimentally determined FAD signal was a superposition of the different layers of the slice, the integral signal (red line) shifted to a more reduced state when oxygenation decreased from 95% to 50%. The hypoxic regions increased when an electrical stimulus was applied to the slice under 50% O_2_ (Figure 4). Due to the increased O_2_ consumption, more layers experienced hypoxic conditions. While the superficial, well oxygenated layers responded to the stimulation just as in the case of 95% oxygenation, the lower layers already showed an increased FAD reduction state, which was not increased by electrical stimulation. Similarly, aggravation of the additional ATP demand did not result in an oxidative shift. Therefore, less layers responded to the stimulation (as also observed in experimental data concerning excitability) and the integrated response was decreased in both the oxidation and reduction phase.

The mean relative changes in FAD signal of the different enzyme complexes during electrical stimulation at 95% and 50% oxygenation is depicted in Figure 4. As can be seen, a lower pO_2_ led to a relative decrease in the peak and undershoot component of the FAD transients for the enzymes PDHC and KGDHC, resulting in a similar signal pattern as experimentally observed. The overshoot component was also decreased for SUCCDH, but it showed an increased undershoot component. For FAD associated to the G3PDH, no difference in the response was predicted. Thus, assuming that all FAD moieties contributed equally to the integral experimental signal, our simulations suggested that the FADs associated with the SUCCDH and the G3PDH might not be the dominant signal sources, which is in line with previous findings [39].

### 2.3. FAD Dynamics during Terminal Hypoxia In Vivo

To explore changes in FAD signals in vivo during systemic hypoxia followed by global ischemia, FAD fluorescence was measured at the cortical surface before and during euthanasia. For this purpose, three animals anesthetized with isoflurane (~2.5%) were killed by decreasing the fractional inspiratory O_2_ from 50% to 0% (i.e., 100% N_2_) in the ventilator air (Figure 5 and Appendix A). FAD imaging, ECoG and [K^+^]_o_ were simultaneously recorded through a small cranial window in the frontal cortex (see Methods). End tidal CO_2_ (ET_CO2_) and blood pressure were monitored to assess changes in ventilation and systemic circulation (see Methods and Figure 5). After a short period of control, hypoxia was induced and changes in all recorded parameters were evident after ~30 s. Typically, ECoG activity ceased with isoelectricity (so called non-spreading depression of activity [13]), while [K^+^]_o_ steadily increased to 12.4 ± 2.3 mM (mean ± standard deviation, *n* = 3) until terminal depolarization occurred. In the early phase of hypoxia, FAD fluorescence displayed a clear reductive shift in all recordings (to ~85%) while ET_CO2_ and systemic blood pressure started to decrease. Importantly, FAD reduction, increase in [K^+^]_o_ and ECoG isoelectricity were the earliest signs of beginning hypoxia with latencies within the 1^st^ minute after starting the ventilation with pure N_2_ (34.8 ± 8.3 s, 34.2 ± 7.2 s and 40.8 ± 21.5 s for FAD, [K^+^]_o_ and isoelectricity respectively, *n* = 3). During persisting hypoxia, cardiorespiratory arrest (lowest values of ET_CO2_ and blood pressure) preceded terminal SD observed as a sharp negative direct current (DC) shift and a simultaneous sharp increase of [K^+^]_o_ to 44.1 ± 11.1 mM (latency of 160 ± 22 s from commencement of hypoxia). During terminal SD, the FAD signal displayed a final oxidative peak. Although FAD reduction occurred earlier than blood pressure decay, we asked whether erythrocytes and blood flow contributed to fluorescence changes as described previously for NADH fluorescence [35]. We performed imaging of FAD prior to hypoxia in two animals using a CCD camera to image vessels and red blood cells. As shown in Figure 5 and in line with previous notions, erythrocytes absent of mitochondria do not contribute to FAD fluorescence [44], however further studies are necessary to exclude alterations due to a decrease in rCBF.

### 2.4. Computational Modeling of FAD Response to Hypoxia In Vivo

For modeling of in vivo FAD fluorescence, we used a tissue model describing concentric layers of neuronal tissue around a perfused blood vessel. Tissue depth was assumed to be 35 µm in line with the intercapillary distance in the rodent brain [45]. Glucose, lactate and O_2_ are transported within the vessel by convective transport and by diffusion between vessel and the tissue as well as within the tissue. Similar to our in vitro simulations, the FAD response was assumed to be the integral response of different layers (around the vessels in vivo). Hypoxia induced by ventilation with 100% N_2_ was simulated by decreasing the O_2_ concentration in the blood from 30 mmHg, corresponding to a normal pO_2_ in the brain parenchyma, to 0 mmHg. The global ischemia-induced terminal SD was simulated by simply putting vessel perfusion to a halt. In contrast to the model, arrest of the circulation in vivo initially leads to a gradual decrease in tissue glucose, which is slower than expected from the fall in perfusion, as the cells first use up their energy stores [46]. The terminal SD either in animals or in humans then starts in vivo with a significant delay of several tens of seconds after the critical fall in brain perfusion when ATP has reached a critically low level that is insufficient for fueling the sodium pump [8,13]. Terminal SD shows an abrupt onset at one or more spots in the tissue from where it spreads. During the explosive onset of SD, abrupt release of approximately 90% of the Gibbs free energy contained in the ion concentration gradients across the neuronal membranes occurs [47]. The released energy is converted into heat, which has been recorded with a sensitive thermal detector in the isolated retinae of the bullfrog and toad as a wave of brain temperature increase by about 5 to 30 mK [48]. Accordingly, the SD causes a further drop in the tissue energy stores, as they are now fully used up in an attempt to reestablish the normal transmembrane ion gradients [10,49,50]. In our model, we neglected the initial phase of global ischemia with a mild reduction in cerebral glucose and focused on the later phase when SD on top of the global ischemia induces a further severe reduction in the cellular energy stores. Figure 6 shows the simulated FAD signals in response to hypoxia and terminal SD. Following our in vivo results, we decreased the O_2_ concentration steeply at t = 60 s, causing a short oxidative shift followed by a severe reductive shift in FAD bound to the PDHC and the KGDHC, while FAD bound to the SUCCDH and the G3PDH exhibited only reductive shifts. During terminal SD (see Figure 6Ab), PDHC as well as KGDHC bound FAD re-oxidized before being reduced again (Figure 6Ba,b). In contrast, FAD bound to SUCCDH stayed almost completely reduced, while FAD bound to mitochondrial G3PDH re-oxidized (Figure 6Bc,d). The re-oxidation of mitochondrial G3PDH arises from the depletion of cytosolic NADH as a result of glucose deprivation due to terminal SD, which is a substrate of mitochondrial G3PDH. The persistent reduction of FAD bound to the SUCCDH arises from its coupling to ubiquinone, an electron carrier of the respiratory chain which remains reduced due to lack of O_2_ as the final electron acceptor. The transient re-oxidation of the PDHC and the KGDHC after depletion of glucose comes from the concomitant lack of pyruvate for fueling of the citric acid cycle and represents a state of metabolic collapse due to substrate depletion. Again, the modeled FAD alterations are in agreement with the measured FAD fluorescence, showing an initial decrease induced by hypoxia followed by a transient increase during terminal SD, assuming that the PDHC and the KGDHC are the dominant contributors to the overall signal.

## 3. Discussion

In the present study, we characterized changes in FAD fluorescence and tissue physiology during O_2_ decrease in brain tissue in vitro and in vivo. By combining experimental results with computational modeling, we identified PDHC and the KGDHC as the dominant contributions to the overall FAD signal in the neocortex both in brain slices and in vivo. The individual components of the experimental traces represented individual steps of the hypoxia-associated disturbances in neuronal energy metabolism.

### 3.1. A FAD Reductive Shift Is an Early Sign of Oxygen Depletion In Vitro

In vitro, mild hypoxia was related to a clear and fast reductive shift of FAD (Figure 1). Thus, FAD fluorescence was highly sensitive to indicate the development of hypoxia. The reductive shift was in line with several previous studies [15,51,52]. Furthermore, the changes in FAD redox state related to early hypoxia were tightly coupled to the alteration in [K^+^]_o_ and synaptic transmission. Since synaptic processes and sodium pump activity account for more than 50% ATP consumption in neurons [53,54], a relative lack of ATP could explain the observed synaptic depression and [K^+^]_o_ alterations. Yet, several other mechanisms have also been proposed to mediate hypoxia-associated synaptic depression, including (1) profound alterations in vesicular transmitter release [55,56], (2) activation of ATP-sensitive or G protein-dependent calcium-sensitive K^+^ channels [57,58] and/or (3) release of adenosine by astrocytes [59]. It is assumed that the initial hypoxia-induced increase of [K^+^]_o_ before the occurrence of SD results in particular from the activation of G protein-dependent calcium-sensitive K^+^ channels [56,57]. In the future, further investigations are necessary to understand the causal connection between the observed reductive mitochondrial redox state and the functional changes in neurotransmission during ongoing hypoxia.

To better understand the signal sources of FAD fluorescence, we simulated neuronal metabolism in silico taking into account the laminar profile of O_2_ distribution present in our in vitro preparation (Figure 3 and Figure 4). In line with our experiments on slices, the simulations showed that hypoxia induced a reductive shift in all FAD containing enzymes (i.e., PDHC, KGDHC, G3PDH and SUCCDH) and that the magnitude of the shift increased with the degree of hypoxia (Figure 3). Following the model, the chain of events includes: (1st) hypoxia-associated inhibition of the complex IV of the respiratory chain, (2nd) subsequent reduction of cytochrome C, which in turn limits the activity of complex III (3rd) reducing ubiquinone (4th) and provoking inhibition of complex I (5th). Concerning the different enzymes containing FAD, the model predicted FAD reduction related to lower transfer of electrons between SUCCDH-FAD to ubiquinone (i.e., due to reduction) and lower activity of complex I with subsequent increase in mitochondrial NADH, which in turn slows the electron transfer from FAD bound to PDHC, KGDHC and G3PDH to mitochondrial NAD as acceptor. At downfall of oxidative phosphorylation increases glycolytic activity leading to an increase in cytosolic NADH and consecutive reduction of G3PDH-associated FAD.

Since our slice experiments and theoretical modeling supported the idea of monitoring FAD fluorescence for early recognition of brain hypoxia, we asked whether FAD imaging would be relevant for hypoxia in vivo.

### 3.2. FAD Fluorescence during Hypoxia and Global Ischemia In Vivo

While neuronal NADH fluorescence has been well characterized and established to recognize hypoxia in vivo [34,60,61], there are few studies concerning FAD imaging in vivo in the brain [28,30]. Concerning hypoxia and in line with our results, FAD reduction was measured in the liver [42] but studies addressing changes in FAD fluorescence during hypoxia in the brain are lacking in the literature.

Our measurements in the frontal cortex of Wistar rats replicated our in vitro observation that hypoxia initially generates a reductive shift in FAD fluorescence (Figure 5 and Appendix A). In terms of temporal resolution, FAD reduction due to decreasing O_2_ occurred in the first minute after beginning ventilation with 100% N_2_ and abolishment of ECoG activity and changes in [K^+^]_o_ were almost simultaneous. Importantly, the initial FAD reduction was followed by a gradual recovery towards oxidation. The terminal SD then led to a sharp oxidative FAD peak that remained below baseline followed by renewed FAD reduction.

While substrate availability was stable in vitro, the situation is much more complicated in the in vivo setting where substrate availability depends on the concentration of nutrients and O_2_ in the blood as well as on rCBF. To better understand the FAD signal in vivo, we simulated the FAD fluorescence of neuronal tissue around a central blood vessel supplying glucose and O_2_ (Appendix A). Thereby, we neglected the initial phase of global ischemia with mild reduction in cerebral glucose and focused on the later phase several tens of seconds after the critical drop in brain perfusion when SD suddenly erupts on top of the global ischemia and causes a further decline in the tissue energy stores [8,13]. As in the in vivo situation, there was an initial drop in the FAD signal due to hypoxia when the oxygenation of the blood decreased and a second response when terminal SD occurred. While the first phase was analogous to the in vitro situation, with O_2_ depletion leading to FAD reduction in all FAD-containing enzymes, terminal SD and subsequent glucose deprivation induced non-uniform FAD signals in the different enzymes. As we pointed out earlier, G3PDH is directly coupled to cytosolic NADH. Glucose deprivation leads to a significant decrease in cytosolic NADH, leading to an oxidation of G3PDH-bound FAD. As the cytosolic NAD reduction state is coupled to the mitochondrial NAD reduction state by the malate-aspartate shuttle, glucose deprivation also leads to oxidation of the mitochondrial NAD pool and the directly coupled FAD moieties of PDHC and KGDHC. However, this re-oxidation is only transient, as O_2_ shortage prevails the flux of electrons in the respiratory chain and pyruvate as substrate for the citric acid cycle is still available. That the re-oxygenation of PDHC- and KGDHC-bound FAD does not imply respiratory chain activity can also been seen from the fact that FAD bound to SUCCDH, which directly couples to ubiquinone of the respiratory chain, does not change its reduction state during glucose deprivation (Figure 6).

In our in vivo recordings, we show for the first time that terminal SD during sustained global ischemia strongly correlated with FAD re-oxidation during hypoxia/global ischemia-associated FAD reduction. By integrating the experimental results with predicted metabolic changes in our simulations, it is possible to speculate about the metabolic pathways involved in terminal SD generation and secondary brain lesions.

While the model explained the underlying metabolic mechanism of the FAD signal, it is important to note that the there are other factors in vivo that were not included in the model. In addition to the molecular mechanism outlined above, mitochondrial swelling, transition pore opening, acidification due to excessive lactate production and resulting protein denaturation all may play a role in the integral FAD response. Furthermore, the simple cytoarchitecture employed to simulate the tissue in silico (Appendix A) did not take into account glia and different types of neurons from the pial surface to deeper areas. Further studies are also necessary to image FAD at the different cell types in the neocortex. As SD is the mechanism initiating the cytotoxic edema, it is known to induce profound changes of the tissue’s optical features and this is very likely to have modified the FAD signal [62,63]. These factors might also explain the variance in the signal when comparing different animals. Thus, our establishing work in vivo demonstrated that FAD fluorescence recording to monitor hypoxic insult is technically possible but further studies are necessary to bring this technic from the bench to the bedside. In addition, weak autofluorescence, phototoxicity and photodecomposition are challenging during FAD recording [33]. For invasive in situ recordings, robust miniaturized systems have been developed [62] but biocompatibility issues need clarification since FAD illumination could affect cell function itself [33]. However, importantly for the clinical application, hypoxia, which is the uniform first event in the potentially lethal cascade, is always reflected in a steep decrease in the FAD signal. Thus, FAD fluorescence imaging could potentially enhance invasive neuromonitoring in neurocritical care in the future.

## 4. Materials and Methods

### 4.1. Slice Preparation and Maintenance

For in vitro experiments, seven adult male Wistar rats (~8 weeks) were sacrificed in accordance with the Helsinki declaration and the approval of the LAGeSo Berlin (T0096/02). Animals were decapitated under anesthesia with isoflurane (1.5%) and laughing gas (N_2_O, 70%). Brain slices were prepared as previously described [63]. Artificial cerebrospinal fluid contained (in mM): 129 NaCl, 21 NaHCO_3_, 10 glucose, 3 KCl, 1.25 NaH_2_PO4, 1.6 CaCl_2_, and 1.8 MgCl_2_. Osmolarity and pH were 295–305 mosmol/L and 7.35–7.45, respectively. FAD autofluorescence was performed in a submerged chamber (flow rate 8 mL/min, temperature ca. 34–35 °C) following a recovery period of 2 h in interface conditions.

### 4.2. Electrophysiology, Oxygen Recordings and Fluorescence Recordings in Brain Slices

All experiments were performed in slices of the frontal cortex. Electrical stimulation was applied with a platinum bipolar electrode in the white matter adjacent to the cortex. To check changes in synaptic transmission, single pulses (100 µs duration) were applied during each experimental condition (i.e., 95% and 50% oxygen). Simultaneous field potential (f.p.) and extracellular potassium [K^+^]_o_ measurements were performed in the layer II using double-barreled ion-sensitive microelectrodes constructed as described previously [32]. To induce transient neuronal activation for FAD, pO_2_ and [K^+^]_o_ recordings, 2-s long 20 Hz stimulation trains were applied (single pulse duration 100 µs, interval 50 ms, 40 pulses) in the control condition and during hypoxia (see Figure 1). The pO_2_ was measured using Clark-style oxygen sensors (tip: 10 μm; Unisense, Aarhus, Denmark). Oxygen electrodes were polarized overnight and calibrated before each recording session. Monitoring of mitochondrial metabolism was achieved by measuring the tissue FAD fluorescence in layer II of the temporal cortex with a 20 × 0.5 W objective using a custom-built imaging setup equipped with a light emitting diode (LED, 460 nm wavelength) and a photomultiplier tube (PMT, Seefelder Messtechnik). The LED (Lumen, Prior scientific, Seefelder, Germany) intensity was set to 18%. To reduce bleaching and phototoxicity, we performed excitation with pulsed light as described previously [33]. Slices were perfused with aCSF gassed with carbogen (95% O_2_ ans 5% CO_2_) achieving a tissue pO_2_ of ~ 100 mmHg at ca. 50 µm depth. Hypoxia was induced by reducing the fraction of O_2_ to 50%, while CO_2_ was maintained at 5%.

### 4.3. In Vivo FAD Recordings

The reporting of animal experiments complies with the ARRIVE Guidelines. In vivo pilot FAD recordings were performed in the frontal cortex of three adult Wistar rats (male, age: ~8 weeks) in accordance with the Helsinki declaration and institutional guidelines (LAGeSo, G0264/14). Rats were first anesthetized with isoflurane and laughing gas (induction with 3% and 70% respectively) in an induction chamber. After induction, maintenance was established with a nares-mask and application of 1–2% isoflurane and 50% oxygen. During the whole procedure, pulse oximetry was controlled using a MouseOxplus^®®^ (Starr life Sciences, Oakmont, PA, USA). Analgesia was performed by local infiltration with lidocaine (1%, Braun, Germany) in the ventral cervical region (tracheotomy area), the head (craniotomy) and the pelvic region (arterial cannulation). After tracheotomy, ventilation with a Harvard Small Animal Ventilator Model 683 (Holliston, MA, USA) was commenced (ventilation was always performed with a gas mixture containing fractional inspired oxygen (FiO_2_) of 50%). End tidal carbon dioxide (ET_CO2_) was monitored during the whole experiment and maintained at ~35 mmHg. Arterial cannulation (A. femoralis) was performed to monitor blood pressure. A right mini craniotomy (ca. 1 × 3 mm) 1.5 mm from the middle line at the coronal suture was performed after fixation in a stereotactic system. A chamber was formed with bone cement around the craniotomy for aCSF perfusion. Prior to study of FAD signals during terminal hypoxia, data concerning the effects of anesthesia depth in extracellular potassium homeostasis was recorded (Liotta et al. 2020, in preparation). After dura incision, animals were placed in the imaging system. A double-barreled potassium sensitive microelectrode was placed in the cortex at 50 µm depth to [K^+^]_o_ and electrocorticography (ECoG). FAD was imaged using a photo multiplier with the same configurations as described previously. In two experiments, FAD pictures in the area of interest were taken using a RedShirt NeuroCCD-SMQ camera (Life Imaging Services, Reinach, Switzerland) to differentiate vessels/blood from cortical tissue (see Figure 3). Euthanasia was achieved by ventilation with 100% nitrogen until exitus due to systemic hypoxia.

### 4.4. Data Analysis of Experimental Data

We planned this exploratory hypothesis-generating study with the aim to characterize FAD changes during oxygen deprivation and propose FAD as a possible biomarker to directly monitor neuronal oxidative phosphorylation. Data is described with median and 25th and 75th percentile in brackets or as indicated in the text. Analog signals were digitalized with Power1401 and recorded with Spike2 (Cambridge Electronic Design Limited, Cambridge, UK). Data analysis and statistics were performed using Spike2 and Origin software (Version 6, Microcal Software, Northampton, MA, USA). Fluorescence is shown as Δf/f_0_, where f_0_ is the baseline fluorescence intensity measured 15 s before baseline FAD changes or stimulus FAD signals were analyzed. For statistical inference, we performed paired Student’s *t*-tests. Changes were stipulated to be significant for *p* < 0.05.

### 4.5. Metabolic Model

Stimulus induced FAD transients in slice preparations were simulated using a kinetic model of neuronal energy metabolism [39]. The biochemical and biophysical processes that were included in the kinetic model are depicted in Appendix A. The model describes the molecular resolved central ATP-producing pathways. It distinguishes the cytosolic and mitochondrial compartment and comprises glycolysis, the citric acid cycle, the respiratory chain, oxidative phosphorylation, mitochondrial electrophysiology (including mitochondrial calcium dynamics) as well as the malate-aspartate shuttle and the glycerol-3-phsophate shuttle, coupling the cytosolic and mitochondrial NAD/NADH pools. It describes the exchange of the nutrients glucose, lactate and oxygen with the extracellular compartment. Kinetic rate equations for the individual enzymes were constructed on the basis of kinetic data. For the FAD containing enzymes pyruvate dehydrogenase, α-ketoglutarate dehydrogenase, glycerol-3-phosphate dehydrogenase and succinate dehydrogenase, the rate equation was modeled in two elementary steps: the reduction of FAD to FADH and the subsequent oxidation of FADH to FAD. The metabolic model has been used for the assessment of NADH and FAD fluorescence under various conditions [39,40,41,45]. All rate equations are given in [39].

### 4.6. In Vitro Slice Model

We modeled the slice as a one-dimensional row of diffusively coupled cells. Each of these cells represents the neurons residing in a cell layer parallel to the slice surface. As in [39], the slice model encompasses 15 cells, whereby cell #1 resides at the surface and cell #15 resides at the core of the slice. Each layer was equipped with the metabolic model described above and it was assumed that the specific activity of all metabolic enzymes was equal in each layer. As oxygen reaches the slice from the surface and is diffusively distributed throughout the slice and continuously utilized within the different layers of the slice, an oxygen concentration gradient exists between the surface and the slice core, so that cells in different layers of the slice experience different oxygen concentrations leading to spatially inhomogeneous oxygen availability. In contrast, glucose is assumed to be unlimited in each layer of the slice as the bathing solution of 10 mM greatly exceeds the affinity of the glucose transporter.

### 4.7. In Vivo Tissue Model

We modeled the tissue as 10 concentric layers of neuronal cells surrounding a central supporting blood vessel. The tissue thickness was assumed to be 35 µm corresponding to the average inter-capillary distance of rodent brain [45]. Again, each layer was equipped with the metabolic model described above and it was assumed that the specific activity of all metabolic enzymes was equal in each layer. We modeled oxygen and glucose supply by diffusive transport between vessel and the first layer and by diffusion between the layers. It was assumed that the outflow of blood from the vessel was equal to the inflow of blood into the vessel in all conditions so that no accumulation of blood in the considered region occurs. Partial oxygen pressure in the blood was set at 50 mmHg and glucose concentration was set as 2.4 mM under standard conditions.

For technical details and all rate equations see [39]. All simulations were made using MATLAB Release2012a (The MathWorks, Inc., Natick, MA, USA).

## Figures and Tables

**Figure 1 ijms-21-03977-f001:**
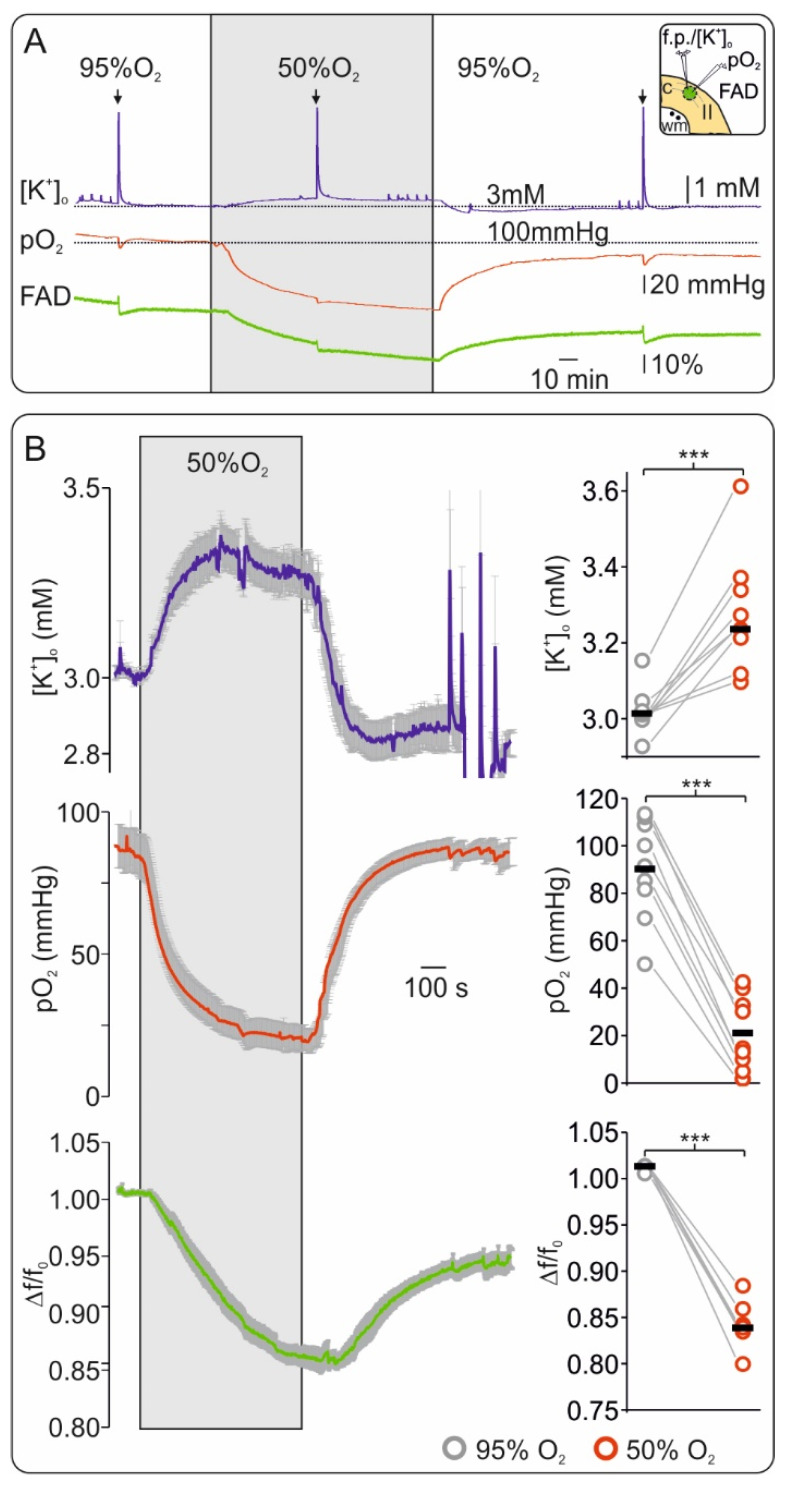
Flavin adenine dinucleotide (FAD) fluorescence is highly sensitive to short hypoxia in vitro. (**A**) Exemplary experiment of simultaneous recording of extracellular potassium ([K^+^]_o_) (upper trace blue), tissue oxygenation (pO_2_) (middle trace, red) and FAD (bottom trace, green) showing the effects of short hypoxia (application of 50% oxygen, grey surface) and reoxygenation. Decreasing O_2_ supply generated a slight increase in interstitial K^+^ and a marked reductive shift of FAD fluorescence. Inflections in the traces resulted from electrical stimulation (arrows on top represent stimulation events of 20 Hz during 2 s, see Figure 2). Thus, the FAD signal was highly sensitive and timely coupled to changes in O_2_ supply. Right on top: graphical representation of cortical slice detail and recording technic (c: neocortex, wm: white matter, II: layer II). Ion sensitive electrode and O_2_ electrode were positioned in layer II of the frontal cortex in the area of interest for FAD imaging (green circle) while electrical stimulation was performed in the adjacent white matter (black dots). (**B**) Left: Average curves (with standard errors) for all experiments concerning changes in [K^+^]_o_ (top, blue line), pO_2_ (middle, red line) and FAD (bottom, green line). Right: statistic plots of single values for each experiment (grey circles: control, red circles: hypoxia, black lines: median values). All parameters were recorded simultaneously, *n* = 9, *** = *p* < 0.001.

**Figure 2 ijms-21-03977-f002:**
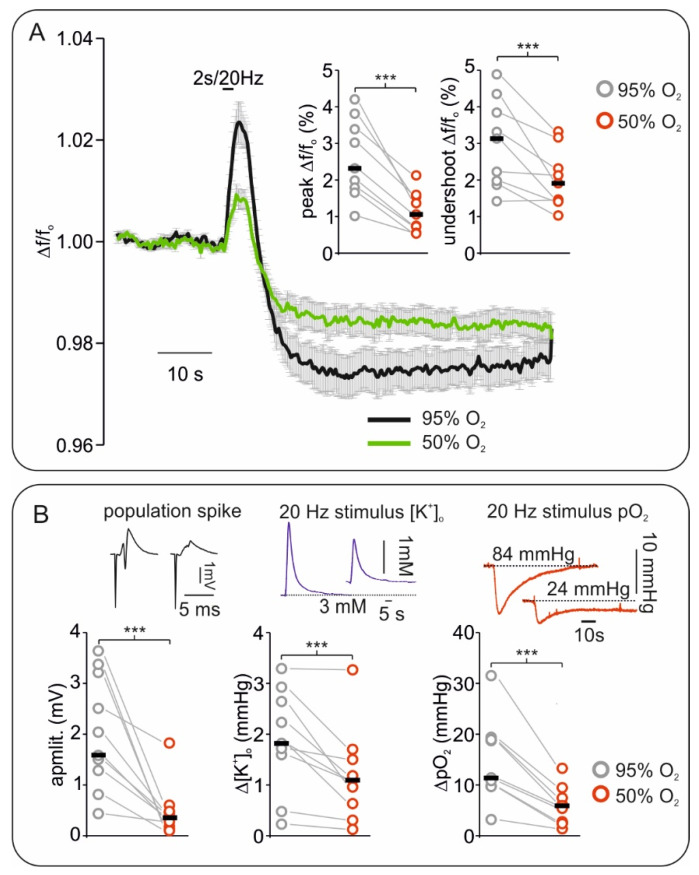
Effects of hypoxia on stimulus induced Flavin adenine dinucleotide (FAD) signals correlates with changes in electrophysiology. (**A**) Averaged curve of stimulus induced FAD transients elicited by 20 Hz/2 s stimulus train. Compared with the control (black curve), peak (oxidation) and undershoot (reduction) FAD response to stimulation decreased during hypoxia (green curve, error bars represent standard error). Right corner on top: plot of changes in FAD oxidative peaks and reductive undershoots during 95% and 50% oxygenation (grey circles: control, and red circles: hypoxia, black lines: median values). (**B**) Plots and exemplary traces of stimulus induced field potential, extracellular potassium ([K^+^]_o_) and tissue oxygenation (pO_2_) changes during hypoxia. Left, the amplitude of single pulse stimulus-induced population spikes (example on top) decreased during hypoxia suggesting impaired synaptic transmission. Middle: [K^+^]_o_ increases elicited by 20 Hz tetanus decreased in amplitude under hypoxia. Right: pO_2_ dips simultaneously recorded with tetanus-induced FAD and [K^+^]_o_ changes decreased. Plots: grey circles: control, red circles: hypoxia, and black lines: median values. All parameters were recorded simultaneously, *n* = 9, *** = *p* < 0.001.

**Figure 3 ijms-21-03977-f003:**
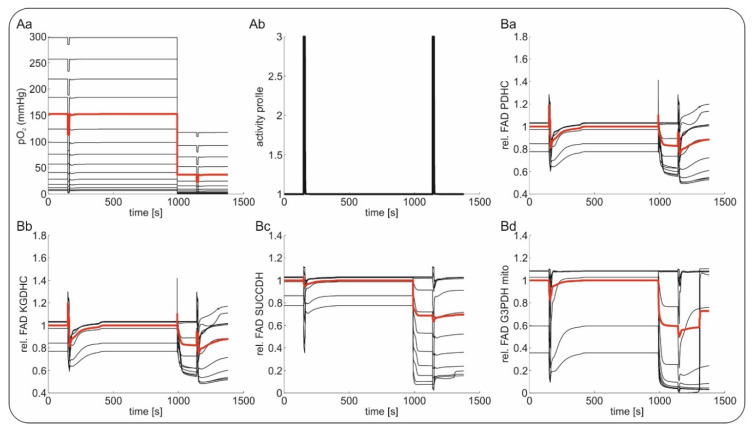
In silico reconstruction of the Flavin adenine dinucleotide (FAD) redox state during hypoxia in brain slices. (**Aa**) Reconstruction of tissue oxygenation (O_2_) gradients in a brain slice in experimental conditions under 95% and 50% oxygenation. (**Ab**) Similar to the increased energy demand elicited experimentally by electrical stimulation trains (20 Hz/2 s), periods of increased neuronal activation were simulated in silico by increasing intracellular Ca^2+^, which generated an activity-dependent drop in intracellular ATP. (**Ba**–**d**). Simulations of changes in FAD fluorescence in different layers during 95% and 50% O_2_ of pyruvate dehydrogenase (PDHC, (**Ba**)), α-ketoglutarate dehydrogenase (KGDHC, (**Bb**)), succinate dehydrogenase (SUCCDH, (**Bc**)) and mitochondrial glycereol-3-phophate dehydrogenase (G3PDHmito, (**Bd**)). Black lines depict the different layers of the slice; the red line gives the mean signal of all black lines.

**Figure 4 ijms-21-03977-f004:**
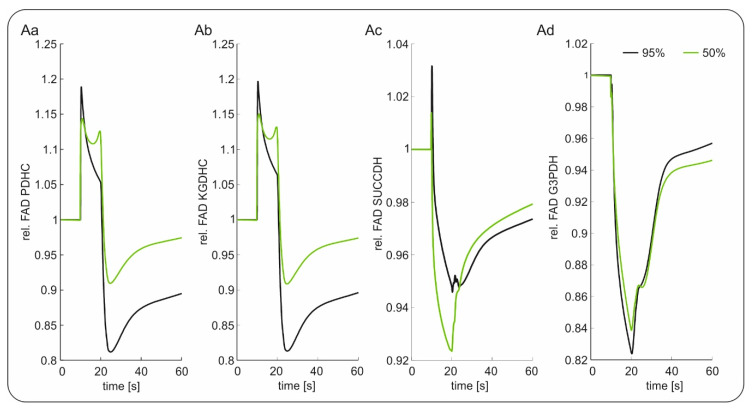
Modeling of relative changes in Flavin adenine dinucleotide (FAD) signaling during electrical stimulation with 95% oxygenation (black) and 50% oxygenation (green); (**Aa**) FAD signal of pyruvate dehydrogenase (PDHC); (**Ab**) FAD signal of α-ketogluterate dehydrogenase (KGDHC); (**Ac**) FAD signal of succinate dehydrogenase (SUCCDH); (**Ad**) FAD signal of mitochondrial glycereol-3-phophate dehydrogenase (G3PDH).

**Figure 5 ijms-21-03977-f005:**
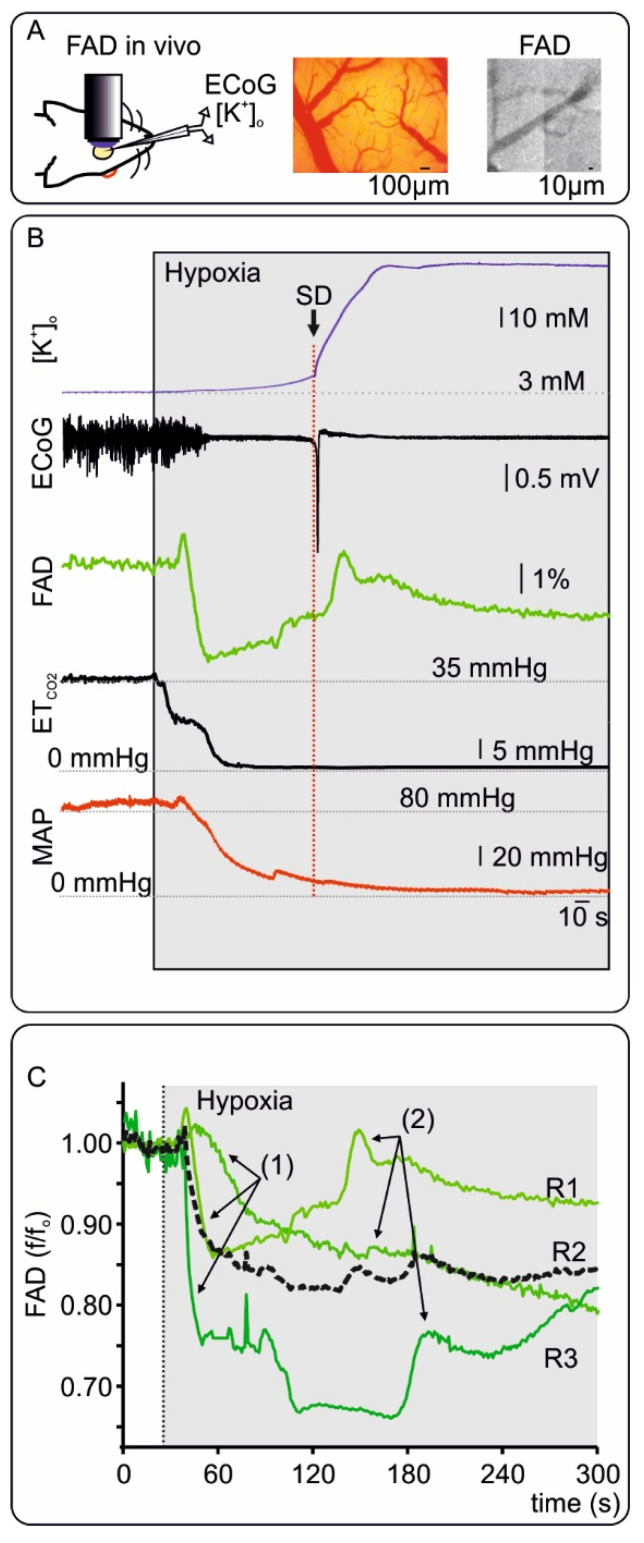
Changes in Flavin adenine dinucleotide (FAD) fluorescence in vivo during hypoxia. (**A**) Left: Representation of in vivo FAD microscopy combined with alternate current (AC) electrocorticography (ECoG) and extracellular potassium ([K^+^]_o_) measurements. Middle: Picture of the cortex with pial vessels. Right: CCD picture (460nm LED illumination and FAD filter) showing the cortical surface emitting fluorescence and dark pial vessels. (**B**) Exemplary experiment concerning in vivo changes in FAD during terminal hypoxia and related signals. A few seconds after beginning ventilation with 0% oxygen, [K^+^]_o_ (blue, trace on top) gradually increased and cortical activity ceased (=non-spreading depression of activity, AC-ECoG trace). FAD-signal displayed a short oxidative peak followed by a marked reductive course. End tidal CO_2_ (ET_CO2_) and mean arterial pressure (MAP) decreased indicating circulatory arrest. After about 110 s of hypoxia and significantly after the circulatory arrest (defined as a fall of ET_CO2_ to zero and a MAP below 30 mmHg), terminal SD then occurred characterized by the sharp and large rise in [K^+^]_o_ and parallel abrupt large negative shift of the direct current (DC)-ECoG. The terminal SD led to a last oxidative FAD-Peak. (**C**) FAD traces from three in vivo FAD measurements (R1-3) during hypoxia. Black dotted line represents the average curve of all traces. As shown previously, acute hypoxia is related to FAD reductive shift (1) and an oxidative peak when terminal SD occurs (2). The exemplary recording in B correspond to R1 in C. For details concerning the individual recordings, see Appendix A.

**Figure 6 ijms-21-03977-f006:**
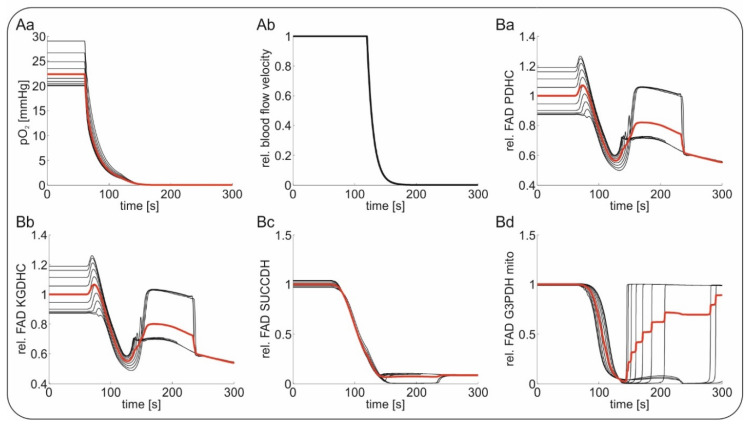
Stimulation of in vivo Flavin adenine dinucleotide (FAD) fluorescence during hypoxia and subsequent global ischemia-induced terminal spreading depolarization (SD); capillary tissue oxygenation (pO_2_) was decreased from 30 mmHg to 0 mmHg at *t* = 60 s followed by further substrate depletion during terminal SD. Red traces indicate average values of the black traces indicating the different layers around the vessel. (**Aa**) The course of the decrease in oxygen during simulated hypoxia mimicking in vivo experiments; (**Ab**) followed by maintained hypoxia, the relative blood flow velocity in the tissue was stopped, simulating the events of global ischemia and terminal SD; simulated relative changes signal of FAD bound to pyruvate dehydrogenase (PDHC, (**Ba**)); to α-ketogluterate dehydrogenase (KGDHC, (**Bb**)); to succinate dehydrogenase (SUCCDH, (**Bc**)) and to mitochondrial glycereol-3-phophate dehydrogenase (G3PDH, (**Bd**)).

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
