# Peer review of "Flavin Adenine Dinucleotide Fluorescence as an Early Marker of Mitochondrial Impairment During Brain Hypoxia"

_ijms, 2020, doi:10.3390/ijms21113977_

Round 1

Reviewer 1 Report

The main problem is that the target tissue oxygen level is not known, so a quantitative correlation between tissue oxygen and FAD reductive shift cannot be assessed. Hypoxia is used as a general term in several instances.

Author Response

Thank you very much for this comment. In this work we described that a reductive shift of FAD characterized the situation of oxygen depletion in both in vitro and in vivo. In the in vitro part of the study, oxygen levels are lowered to values which were not always in the range of hypoxia. However, FAD reductive shift always happened suggesting that also early oxygen depletion can reliably be monitored with FAD fluorescence. Concerning the absolute correlation between FAD and tissue pO2, we are still working to improve FAD imaging and correlate it changes with other signals (i.e. pO2, regional cerebral blood flow, interstitial ions). For this work, we established in vivo FAD measurement so now we will be able to start more precise measurements and quantitative analysis of the signal during different physiological and pathological situations.  

Reviewer 2 Report

The manuscript by Berndt et al., describes the fluorescent changes in FAD following hypoxia in the brain. Authors conclude that FAD fluorescence can be used to monitor mitochondrial function during hypoxia and therefore has potential as a diagnostic or brain monitoring tool. Overall an exceptionally well written and succinct manuscript with a clear logical flow. The rationale behind the study is solid and the topic is of significant interest to scientists and clinicians. The introduction provides sufficient review of literature and offers contrast of current fluorometric approaches for monitoring brain injury in a critical care setting. The findings of the first experiments (2.1) are particularly exciting as the authors demonstrate a clear simultaneous decrease in FAD fluorescence in response to hypoxia, indicating a reductive shift in mitochondrial complexes. The conclusions drawn from the study are fair and have been well justified without over-interpreting or extending the reach of data.

My primary comment is regarding the computational models. A key limitation is the oversimplification of complex neuronal networks and the difficulty to incorporate compensatory mechanisms that may be involved under pathological conditions. Here the authors have explicitly stated all assumptions and validly justified their approaches. However, I would suggest the authors address a few of the limitations in more depth within the discussion. In particular, the significant assumption of the model presented here is that each layer displays homogenous neuronal distribution and will therefore require the same energy demands, and consumption. I feel these assumptions strongly favour the hypothesis rather than necessarily reflecting a true physiological response. Cellular distribution varies greatly (both neurons and glia) from the pial surface to the deeper regions. This variation in abundance will likely be associated with different energy demands and functional capacity of cells present. These distributions will therefore significantly impact FAD fluorescence and are not likely to be uniform as described.

In addition to this, hypoxia rarely causes uniform global injury. Particularly, focal injury of neurons will also alter activity and function of glial cells in proximity, again this will alter energy demands. There is also evidence that glia may modulate neurotransmission under high frequency conditions which again will alter their energy demands. While difficult to model, these interactions are worthy of discussion as they have largely been omitted from the present models and are likely to be key contributors to energy demands.

General query; how specific are the alterations in FAD fluorescence to the neuronal energy expenditure? Is it possible to image changes in specific live cell populations?

Please see below for a few further suggestions that may improve the manuscript. Points are listed in the order of appearance not in the order of importance.

RESULTS:

-Editing detail for Figure 4: The figure number has been omitted from the legend, reads “Figure    .”

-Figure 4 legend correction: For consistency authors should adapt both the figure & legend to state G3PDH. At present both are written as G3DH.

-Figure 5: Figure 5B shows data from ‘exemplary experiment.’ Can the authors indicate in text which rat (R1,2 or 3) this data is from? Would be nice to relate to C, which shows three distinct variations of FAD responses.

OR better still can authors compile a figure to show physiological parameters in a manner like 5C. It would be very interesting to see the changes in parameters for individual rats and how they shape the trajectory of observed FAD levels (potentially as supplementary figure).

Methods:

4.3 In vivo FAD recordings; can the authors state the sex of rats used. For electrophysiological studies the authors used specifically male rats. If a mixed sex sample was used can authors explain why and how this may influence results.

Author Response

We thank to the reviewer for his accurate and motivating remarks.

We include a sentence in the discussion to describe the limitation of the model conderning the cytoarchitecture and the different types of cells involved in the generation of the FAD fluorescence.

Using two photon microscopy to measure FAD, we are able to image mitochondria in hippocampal cultures but time resolution and phototoxicity are limitating factors. We are working to improve our imaging technic using LEDs and better detectors to enhance spatial and temporal resolution. In this paper we show our very first in vivo measurements of FAD so we hope in a near future to be able to image mitochondria in different types of cells in vivo and the changes in FAD fluorescence intensity.

Concerning FAD and oxygen expenditure: at the present, we are only able to calibrate our model with changes in CMRO2 in vitro (see Berndt et al. 2018, PMID: 30143847). In vivo, the situation is much complicated due to changes in regional cerebral blood flow, changes in metabolism due to anaesthesia  and technical issues concerning simultaneous reocrdings of FAD fluorescence, pO2 and elctrophysiology. Our group is preparing a publication including the different aspects involved in FAD fluorescence and metabolic demand under anesthesia. 

Specific points:

-Editing detail for Figure 4: The figure number has been omitted from the legend, reads “Figure    .”

Answer: We corrected this edition mistake. 

-Figure 4 legend correction: For consistency authors should adapt both the figure & legend to state G3PDH. At present both are written as G3DH.

Answer: We corrected the legend

-Figure 5: Figure 5B shows data from ‘exemplary experiment.’ Can the authors indicate in text which rat (R1,2 or 3) this data is from? Would be nice to relate to C, which shows three distinct variations of FAD responses.

Answer: we described at the end of the figure caption that the data in B is R1.

OR better still can authors compile a figure to show physiological parameters in a manner like 5C. It would be very interesting to see the changes in parameters for individual rats and how they shape the trajectory of observed FAD levels (potentially as supplementary figure).

Answer: we add a new supplemetanry figure with all 3 recordings and the changes in electrophysiology, extracellular potassium, systemic blood pressure and end-tidal CO2 (see suppl. fig. 2 )

Methods:

4.3 In vivo FAD recordings; can the authors state the sex of rats used. For electrophysiological studies the authors used specifically male rats. If a mixed sex sample was used can authors explain why and how this may influence results.

Answer: we specified the sex of the rats in the methods. We worked with males.  

Again, we would like to specially thank to the reviewer for his encouraging revision!

Reviewer 3 Report

The paper presents the possibility of using FAD fluorometry to monitor consequences of hypoxia in brain tissue in vitro and in vivo basing on animal model. It was shown that FAD (Flavine adenine dinucleotide) fluorescence could be used to characterize mitochondrial function during O2 decrease because emission of FAD fluorescence is highly sensitive to hypoxia and correlates with changes in electrophysiology. In addition, it should be noted that computational modelling of FAD response to hypoxia was included into analysis of experimental data. By combining experimental results with computational modelling the dominant contributions ( PDHC and KGDHC) to the overall FAD signal in the neocortex both in brain slices and in vivo were found. It follows from this paper that assay of mitochondrial function by means FAD imaging could be a potential diagnostic tool to monitoring metabolic process of brain in patients at risk for ischemia.

I believe that the comprehensive research on FAD as an early marker of mitochondrial impairment during brain hypoxia is worth publishing

However I have some remarks:

  1. In my opinion section “Materials and Methods ” should be before “Results”
  2. Figure caption on page 7 (part 2.2) should be completed. Is it Figure 4?
  3. Is the phrase “Materials and Methods" necessary in the beginning of discussion (part 3.Discussion, page 10, line 16)

Author Response

We thank the reviewer for the positive comments on our work. This is for us always a source of motivation to continue our research.

Specific Points

However I have some remarks:

  1. In my opinion section “Materials and Methods ” should be before “Results”
  • Answer: thank you for this comment. We think that the rationale of a classical paper structure with "Materials and Methods" before the "Results" is appropiate. However, we edit the paper following the author guidelines of MDPI.                                                                                                                                                                                                               2.Figure caption on page 7 (part 2.2) should be completed. Is it Figure 4?
  • we corrected this. is an edition mistake.                                                                                                                                                                        3. Is the phrase “Materials and Methods" necessary in the beginning of discussion (part 3.Discussion, page 10, line 16)
  • we errased this. Another edition mistake.